# Aortic Valve Stenosis and Cancer: Problems of Management

**DOI:** 10.3390/jcm12185804

**Published:** 2023-09-06

**Authors:** Gloria Santangelo, Silvia Moscardelli, Lucia Barbieri, Andrea Faggiano, Stefano Carugo, Pompilio Faggiano

**Affiliations:** 1Department of Cardio-Thoracic-Vascular Area, Foundation IRCCS Cà Granda Ospedale Maggiore Policlinico, 20154 Milan, Italy; gloria.santangelo@policlinico.mi.it (G.S.); lucia.barbieri@policlinico.mi.it (L.B.); andreafaggiano95@gmail.com (A.F.);; 2Division of Cardiology, Department of Health Sciences, San Paolo Hospital, University of Milan, 20133 Milan, Italy; silvia@moscardelli.it; 3Department of Clinical Sciences and Community Health, University of Milan, 20133 Milan, Italy; 4Fondazione Poliambulanza, Cardiothoracic Department Unit, 25100 Brescia, Italy

**Keywords:** aortic stenosis, cancer, valve replacement, cardio-oncology, transcatheter valve implantation, radiation therapy

## Abstract

Aortic valve stenosis and malignancy frequently coexist and share the same risk factors as atherosclerotic disease. Data reporting the prognosis of patients with severe aortic stenosis and cancer are limited. Tailoring the correct and optimal care for cancer patients with severe aortic stenosis is complex. Cancer patients may be further disadvantaged by aortic stenosis if it interferes with their treatment by increasing the risk associated with oncologic surgery and compounding the risks associated with cardiotoxicity and heart failure (HF). Surgical valve replacement, transcatheter valve implantation, balloon valvuloplasty, and medical therapy are possible treatments for aortic valve stenosis, but when malignancy is present, the choice between these options must take into account the stage of cancer and associated treatment, expected outcome, and comorbidities. Physical examination and Doppler echocardiography are critical in the diagnosis and evaluation of aortic stenosis. The current review considers the available data on the association between aortic stenosis and cancer and the therapeutic options.

## 1. Introduction

The coexistence of cancer and calcific aortic valve stenosis (AS) is a common medical scenario, especially in the elderly, due to sharing risk factors (i.e., hypertension, obesity, diabetes, smoking, dyslipidemia), the inflammatory state associated with malignancies, and/or cardiotoxic effects of cancer therapy [1]. As reported in studies listed in Table 1, the prevalence of cancer in patients with severe AS varies between 5.4 and 37.8% [2,3,4]. Data reporting the prognosis of patients with severe AS and cancer are limited. In a 10-year single-center retrospective study, cancer patients with severe AS (mean aortic valve area 1.0 ± 0.3 cm^2^) had a 5 year mortality of 48%; 59% deaths were due to cancer progression, and 31% were due to heart failure (HF) and stroke [5]. Minamino-Muta et al., in a Japanese retrospective study of 3815 patients in a multicenter AS registry, found that outcomes are worse not only in patients with active cancer but also in those with a previous history of malignancy [6]. Mortality was mainly cancer related, with comparable aortic valve-related deaths between cancer and noncancer patients. Despite the increasing prevalence of AS and cancer, death rates have been steadily declining with the introduction of novel therapies [7], but, at present, the optimal strategy for the management of severe AS in patients with an active cancer is unclear. Cancer patients are routinely excluded from clinical trials because of poor long-term prognosis. Active malignancy often hinders the decision to proceed with invasive procedures, such as cardiac surgery. Furthermore, cancer patients have additional risks due to prior exposure to potentially cardiotoxic chemotherapy, prior chest radiation, immunocompromised state, and increased risk of both bleeding and thromboembolic disease [8]. In patients with cancer, AS may interfere with optimal antineoplastic management (i.e., high-risk oncological surgery or potentially cardiotoxic chemotherapies). Symptomatic AS is occasionally diagnosed in cancer patients undergoing cardiovascular evaluation; likewise, cancer is often recognized during assessments preceding aortic valve interventions. In these complex cases, physicians face a difficult treatment decision.

Usually, an echocardiographic evaluation is done before chemotherapy is started; the presence of LV dysfunction before generally represents a risk situation for chemotherapy; on the other hand, LV dysfunction in the presence of severe AS, especially if symptomatic, represents an indication for the treatment of valvular disease soon.

Khrais et al. found [9] that AS can be seen as a prognostic risk factor for adverse outcomes in patients with colorectal cancer due to higher rates of lower gastrointestinal bleeding and resulting iron-deficiency anemia.

Severe symptomatic AS in patients with cancer requires careful assessment in order to select the appropriate therapeutic choices and their timing (i.e., valve treatment first versus cancer treatment first). First of all, cancer localization and therapy, anemia, self reduction of physical activity, etc. may be important confounding factors in the definition of symptomatic vs. asymptomatic AS. If the stenosis is severe, the dosage of Nt-proBNP/BNP can be useful to attribute the genesis of the symptoms to the valve disease. Echocardiography is key to confirming the diagnosis and severity of AS, assessing valve abnormalities, left ventricular (LV) hypertrophy and function, detecting other valve diseases or aortic pathology, and providing prognostic information. If feasible, exercise testing, especially exercise echocardiography, can clarify the nature of symptoms. These echocardiographic findings must be considered together with coronary/vascular diseases and cardiovascular medications. LV systolic dysfunction represents an important prognostic factor and is included in the current operative risk scores [10]. It can be due to long-standing pressure overload, associated aortic regurgitation or mitral valve disease, coronary artery disease, but, also, to cardiotoxicity induced by cancer treatment (especially anthracyclines and targeted therapies such as tyrosine kinase inhibitor, antihuman epidermal growth factor receptor 2, antivascular endothelial growth factor, or proteasome inhibitors) [11]. Ezaz and colleagues [12] developed a risk-factor-scoring tool for patients on trastuzumab to help identify those at highest risk of developing HF or cardiomyopathy. A seven-factor risk (age, adjuvant chemotherapy, coronary artery disease-CAD, atrial fibrillation or flutter, diabetes mellitus, hypertension, and renal failure) score was derived and validated. Low (0–3 points), medium (4–5 points), and high (=6 points) risk strata had three-year rates of HF or LV dysfunction of 16.2%,26%, and 39.5%, respectively. LV dysfunction can remain asymptomatic for a long time [13], but once symptomatic, the prognosis is among the worst in the HF population [14]. Moreover, chest radiation and cardiotoxic drugs (i.e., anthracyclines) have been noted to produce de novo AS via valve leaflet thickening, fibrosis, retraction, and calcification [15], but, at the present, the impact that they may have on AS progression has not been studied. Bravo-Jaimes et al. have found that patients with mild or moderate AS and cancer are more likely to die before having AS progression, which is, in turn, associated with CAD and prevalent cyclophosphamide use [16].

Surgical valve replacement (SAVR), transcatheter valve implantation (TAVR), balloon valvuloplasty, and medical therapy are possible treatments for aortic valve stenosis, but when malignancy is present, the choice between these options must take into account the stage of the cancer and associated treatment, expected outcome, and comorbidities [17]. However, cancer-survivor patients with a confirmed remitted malignancy and evidence of severe AS, after an accurate multidisciplinary team evaluation with oncologists, an interventional cardiologist, and a heart surgeon, can be considered similar candidates to patients with no previous cancer history in terms of eligibility for aortic valve replacement. As recently reported by Płońska-Gościniak, patients with severe, pre-existing cancer and heart-valve disease should be managed according to the 2021 guidelines of the European Society of Cardiology and Cardiothoracic Surgery taking into consideration the cancer prognosis and patient preferences [18]. 

## 2. Pathophysiology

Clinical risk factors associated with AS development and progression mirror those associated with atherosclerosis, and because many are shared by cancer (advanced age, smoking, hypertension, hypercholesterolemia, obesity, metabolic syndrome, diabetes, and elevated lipoprotein (a) levels) prevalence and incidence rates of both disorders are rising simultaneously [19,20]. These common conditions, together with microbial and viral infections, allergen exposure, radiation, toxic chemicals, alcohol consumption, tobacco use, and other chronic and autoimmune diseases, induce inflammation [21]. It is now known that inflammation mediates all atherosclerosis stages, from initiation to progression and, ultimately, plaque unstabilization and thrombosis. Conditions such as hypertension, smoking, dyslipidemia, and insulin resistance all appear to trigger atherosclerosis, by promoting the expression of adhesion molecules by endothelial cells, allowing leukocyte attachment to blood vessel walls that normally resist their attachment. In recent decades, extensive factual and circumstantial evidence has shown several cancer types to be induced by infection or chronic inflammatory disease (e.g., human papillomavirus and cervical cancer, Helicobacter pylori and stomach cancer, and Epstein–Barr virus and lymphoma) [22]. As stated by Koene et al. [21], controlling cardiovascular disease risk factors can help reduce the risk of cancer. There is an urgent need to improve the health status of the population to reduce the prevalence of both diseases.

Although chronic inflammation is an indispensable feature of the pathogenesis and progression of both cardiovascular disease and cancer, additional mechanisms can be found at their intersection, such as nonmodifiable risk factors, including age, sex, and race/ethnicity, which are uncontrollable. There are obvious differences between male and female organs and hormonal fluctuations that influence both cardiovascular disease and cancer progression. Of all nonmodifiable risk factors, age is a steady independent variable with regard to cardiovascular disease and cancer, yet the associations between age and disease onset can be highly influenced by lifestyle parameters, such as diet, physical activity, body mass index, and smoking.

## 3. Treatment

Currently, the optimal strategy for the management of severe AS in patients with an active cancer is still unclear. Tailoring the correct and most optimal care for cancer patients with severe AS is complex. Asymptomatic patients with severe AS, in the absence of adverse prognostic features such as reduced LV ejection fraction, or symptoms appearance during an exercise stress test, are recommended for a watchful waiting approach, with regular and frequent follow-up and prompt intervention in case of clinical progression (i.e., symptoms). In this way, asymptomatic patients can proceed with their antineoplastic therapy without interruptions or delays. Medical treatment of hypertension and hyperlipidemia, according to current international guidelines, is recommended for patients with severe AS. According to the 2020 American College of Cardiology Foundation/American Heart Association guidelines, adult patients with symptomatic severe AS (stage D), or with asymptomatic severe AS with LVEF <50%, or need of other cardiac surgery have an indication for aortic valve replacement (AVR). If the risk for a SAVR is high or prohibitive, decision-making focuses on TAVR or palliative care, depending on the life expectancy [23]. In the 2021 European Society of Cardiology Guidelines for The Management of Valvular Heart Disease, AVR is indicated in patients with symptomatic severe AS, except for those in whom the intervention is unlikely to improve quality of life or survival (due to severe comorbidities) or for those with concomitant conditions associated with survival <1 year [24]. In the past years, in patients with severe AS, priority was given to the treatment of neoplastic disease rather than the treatment of severe valvular disease. However, patients undergoing SAVR have shown markedly better survival, due to better resilience to anemia, infection/sepsis, and rapid volume changes from chemotherapy regimens or hypotension/volume loss during surgical procedures, not uncommon during cancer treatment. It must be said, anyway, that patients with severe AS are not excellent candidates for surgery mainly due to comorbidities that increase the estimated periprocedural morbidity and mortality [2]. Conflicting results come from reports where SAVR is performed before cancer surgery [25]. A fundamental problem of SAVR in cancer patients is that open surgery requires extracorporeal circulation. Among various other systemic effects, cardiopulmonary bypass can cause immunosuppression, increase inflammation (as demonstrated by a significant increase in TNF-alpha, Il-10, Il-6, Il-1, and TGF-beta), and worsen cancer outcomes. So, precisely because of the immunosuppressive effects, patients with hematologic cancers are at risk of having worse outcomes than those with solid tumors and better immune systems. However, the relationship between the use of extracorporeal circulation and cancer progression has not yet been clearly demonstrated. Among the comorbidities, we should also consider the vascular fragility that patients with active cancer can develop, and which can sometimes be caused by anticancer drugs or radiotherapy. In addition, cardiac surgery recovery times are longer, and this could lead to a delay and lengthening of the antineoplastic therapy times.

Even if is mandatory to consider each case individually (SAVR vs. TAVR), it is reasonable to conclude that TAVR, for cancer patients with severe AS, can more frequently be the best clinical choice by avoiding cardiopulmonary bypass and all its consequences. One of the biggest advantages of TAVR is its minimal invasiveness and, therefore, shorter recovery time. Moreover, thanks to the increasing access of cancer patients to TAVR, delays in cancer treatment have been significantly reduced from about 2 months after cardiac surgery to 2 weeks [26]. TAVR does not require a median sternotomy or cardiopulmonary bypass and can be performed under local anesthesia, which reduces the overall time required to complete the procedure, which benefits patients with malignancies. Manger et al. and Landes et al. reported that TAVR periprocedural mortality and major complication rates were equivalent in patients with and without active cancer [3]. Kojima et al. reported no difference in terms of complications between patients undergoing TAVR with and without active cancer [27]. However, despite the technical difficulties for open surgery that may be overcome by TAVR, major comorbidities may influence post-TAVR prognosis just as with SAVR [28]. To be eligible for TAVR, as mentioned before, cancer patients should have a prognosis of 1 year or greater. However, precise estimation of prognosis has always been very difficult in these patients, and even more so lately, thanks to the rapid expansion of new and innovative cancer therapies. A fundamental element to consider in choosing the most suitable type of intervention for the patient, in addition to the prognosis, is the stage of neoplastic pathology. Patients with a history of cancer, who are judged in remission by the oncology team, are usually eligible for TAVR. Patients with early cancer stages, who can safely receive oncologic treatment, could be easily considered for TAVR as soon as remission is confirmed. In other cases, performing TAVR before cancer treatment allows for radical oncologic treatment shortly after valve intervention [29]. Patients with AS and a localized cancer can be stabilized and TAVR can be considered after the exclusion of metastatic disease. Patients with advanced disease stage, metastases, multiple comorbidities, and very short estimated survival may be candidates for balloon valvuloplasty as a “bridge to destination” surgery [30].

In the final stages of neoplastic disease, a more conservative approach aimed at improving quality of life during palliative treatment is preferred. A recent expert consensus issued by the Society for Cardiovascular Angiography and Interventions recommends aortic balloon valvuloplasty or TAVR for cancer patients with AS as either a palliative or cure for valvular disease, to improve quality of life or to facilitate appropriate treatment of cancer therapy. Unfortunately, due to the characteristics of advanced stage cancer patients, it is difficult to conduct large studies, limiting the quality of data to support this approach [30]. A small study from Schechter et al. of 65 cancer patients with severe AS found that valve replacement improves survival, regardless of the type of cancer or anticancer therapy, with TAVR being the most effective [26]. Nowadays, the majority of cancer patients diagnosed with severe AS undergo valve replacement before cancer treatment, with the large majority receiving TAVR more than SAVR. Despite the lower risk of TAVR complications, the literature is not univocal about what are the peri-procedural complications of TAVR that could cause a delay in cancer treatment and modify overall survival. A meta-analysis by Marmagkiolis et al. demonstrates a favorable post-TAVR short-term mortality and remarkable safety, with improved stroke and acute kidney injury (AKI) rates without increased bleeding and the need for new pacemaker implantation in cancer patients compared to controls [7]. Conversely, a meta-analysis from Bendary et al., reported higher rates of postprocedural pacemakers, without any difference in short-term mortality [31]. In a systematic review of Arocutipa et al., AKI occurred more frequently in patients with active cancer [32]. AKI is a very common complication of TAVR and can rate in up to 50% of procedures. Its origin is multifactorial: in addition to the iodinated contrast, bleeding and anemia, volume depletion, microembolisms, hypotension, or nephrotoxic drugs also contribute. Importantly, tumor type also plays a role in the risk of post-TAVR AKI. Thus, the decision to ultimately pursue TAVR is not an easy choice and involves a multidisciplinary and holistic approach to assessing the appropriateness of intervention. Recently, also in light of the study of Ullah et al. [33], which highlighted different outcomes between SAVR and TAVR based on the tumor location, our group proposed a detailed specific decision-making algorithm for the management of symptomatic severe AS in cancer patients, both active and in remission [34]. Specifically, in the case of active cancer, once it is ascertained that cancer-related life expectancy is >1 year, that cancer treatment is not feasible before AS treatment, and that cancer treatment can be delayed for at least 2 months, the decision-making process is comparable to cancer in remission. In this case, evaluation for the presence of high-risk features for SAVR and/or clinical conditions favoring TAVI TAVR is suggested. Where such conditions are not present, the choice between TAVR and SAVR rests in the judgment of the heart team, including the consideration that the tumor site can influence the management strategy and the personal patient choice (Figure 1).

### 3.1. Radiation Therapy and Aortic Stenosis Treatment

An issue that should not be forgotten in the field of AS and cancer disease is the impact of radiotherapy on these patients. Chest radiation (C-XRT) is part of standard treatment protocols for various malignancies (i.e., lymphoma and breast, lung, and esophagus cancer) [35]. Radiation-induced valvular disease involves a degenerative process with early valve retraction resulting initially in regurgitation and, after, thickening and calcification of the valves leading to stenosis. Fibrosis and calcification of the mitral and aortic valves, especially surrounding structures including the annulus, subvalvular apparatus, and the aorto-mitral curtain [35], have been noted in patients who underwent C-XRT. Left-sided valves are most commonly affected probably due to the stimulus of higher pressure flows.

A dose-dependent toxic effect on the heart has been previously demonstrated, and immunobiological studies have shown, specifically, a dose-dependent effect of aortic valve fibrosis. It has been suggested that >30 Gy is considered a high dose of mediastinal radiation [36]. In terms of screening, the International Cardio-Oncology Society recommends obtaining a routine transthoracic echocardiogram in all patients who are planning to undergo thoracic radiation and every 5 years thereafter to screen for radiation-induced valvulopathy [37]. However, this recommendation has yet to be integrated into routine clinical practice. In a retrospective study of Hodgkin disease survivors with and without prior chest radiation, 6 of 49 (12%) patients who underwent C-XRT developed moderate or severe aortic regurgitation, mitral regurgitation, or AS, whereas only 1 of 29 patients without prior chest radiation developed more than mild AS, and 1 more than mild aortic regurgitation [38]. This patient group is plagued by a high mortality and presents unique challenges in surveillance and in balancing the risks and benefits of treatment [39]. Donnellan et al. [40] compared AS patients with prior exposure to C-XRT to a group with similar AS at baseline but no history of irradiation. Although the progression of AS was similar in both groups, significantly more patients in the C-XRT group underwent AVR for the development of symptoms (80% vs. 50%, *p* < 0.001) during a mean follow-up of 3.6 ± 2 years. Despite that, the C-XRT group had significantly higher long-term mortality than the comparison group. The decision-making of treatment modality for radiation-induced AS should be a multidisciplinary decision that is targeted for the patient’s specific characteristics and needs, taking into account the complexity of anatomy and disease history. Patients with a prior history of radiation to the chest are considered to be at high risk for surgery for numerous reasons: the ascending aorta can be markedly calcified (‘porcelain aorta’) making cross-clamping difficult, the frequent need for associated mitral and coronary surgery, and the presence of pulmonary fibrosis, which correlate directly with mortality postoperatively [36]. Furthermore, C-XRT causes mediastinal adhesions and fibrosis that need to be dissected. They increase the risk of bleeding and poor wound healing. The treatment with debridement may increase the cardiopulmonary bypass time. Therefore, a history of prior chest radiation is now included in the STS risk score before a cardiac surgery, given its significant effect on surgical mortality [41]. A recent matched cohort study found that radiation was associated with a statistically significant increase in in-hospital mortality and a 6 year mortality after SAVR, compared with patients without a radiation history [42]. However, 61% of patients were undergoing SAVR with another concomitant procedure. Isolated SAVR has been shown to have better 5 year survival than combined procedures in patients with radiation-induced AS [43]. TAVR is an increasingly performed procedure and may be an important treatment avenue for patients with radiation-induced AS, taking into account the potential complications, such as fistulization and tissue rupture.

An accurate analysis with computed tomography angiography (CTA) evaluating aortic valve characteristics and size, access route, and degree of aortic calcifications for optimal TAVR planning is always mandatory, especially in this very high-risk setting [44]. Zafar et al. [45] showed in a 2020 systematic review and meta-analysis that TAVR was a safe option for patients with radiation-induced AS. Although current guidelines do not recommend TAVR in patients with a life expectancy of less than 1 year, many cancer survivors do not meet this timeline, and even those on active therapy are experiencing continued improvement in survival. There will, therefore, be a growing need to revisit the option and benefit of TAVR in cancer patients [7].

### 3.2. Aortic Stenosis, Coronary Artery Disease (CAD) and Mitral Regurgitation

Concomitant cardiovascular problems, such as CAD or mitral regurgitation (MR) should be promptly assessed before the treatment of AS. A surgical approach with combined valvular intervention and coronary revascularization may represent an extremely high-risk setting in fragile patients, such as cancer ones. Therefore, a percutaneous approach is reasonably the best option to adopt. As reported above, CTA of the aorta and the iliofemoral arteries is crucial for preprocedural planning, and the combination of coronary computed tomography angiography (CCTA) may be useful for the exclusion of concomitant CAD in order to minimize invasive procedures in these high-risk patients [46]. In the case of concomitant significant CAD, American guidelines [23] recommend percutaneous coronary interventions (PCI) before TAVR in the case of left main disease and significant proximal CAD. Instead, European guidelines [24] suggest basing a decision according to the clinical presentation, coronary anatomy, and extent of myocardial risk. PCI concomitant with TAVR is recommended in patients at high risk of coronary obstruction by the prosthetic aortic valve (e.g., ostial lesion, low left main height, or a valve-in-valve implantation) or in patients in whom it is desirable to minimize dual antiplatelet therapy (DAPT) due to bleeding risk [47]. Regarding antithrombotic therapy, the standard treatment after TAVR is aspirin, while patients with an indication for oral anticoagulation therapy or DAPT should receive the specific treatment according to the pre-existent clinical indication without any concern to the valve procedure. This represents a concrete advantage, especially in patients with higher bleeding risk, such as cancer patients [48]. Furthermore, the progressively younger age of patient candidates for TAVR makes the possibility of reaccessing the coronary arteries increasingly important. Thus, further studies on increasing coronary reaccess after TAVR and the best timing of percutaneous coronary interventions in relation to TAVR are necessary.

The reported prevalence of moderate or severe mitral regurgitation (MR) in AS patients eventually undergoing SAVR or TAVR ranged between 19% and 33% [47,49]. Ruel et al. have found that AS patients with a functional MR ≥ 2+ and a left atrial diameter >5 cm, preoperative peak aortic valve gradient <60 mm Hg, or atrial fibrillation have a significantly higher risk of cardiac HF and persistent mitral regurgitation after AVR than other AS patients. Waisbren et al. [50], in their work, support a conservative, tailored approach to concomitant mitral surgery in patients presenting for the correction of AS who demonstrate functional regurgitation. Finally, in patients with severe MR, there is not enough experience to make recommendations about surgery versus combined or sequential TAVR and percutaneous mitral edge-to-edge repair. Currently, no consistent data is available about the coexistence of AS and MR and the related treatment in the subgroup of cancer patients.

### 3.3. Aortic Stenosis and Cardiac Amyloidosis

The prevalence of calcific AS and cardiac amyloidosis (CA) increases with age, and their association is, as expected, not uncommon in the elderly. Deposition of an amyloid substance, especially transthyretin, can involve any cardiovascular structure, including the aortic valve and myocardial walls, and it may contribute to the initiation and progression of AS, as well as progressive myocardial dysfunction. Until now, there is no recommendation or consensus on whether patients with AS should be systematically screened for CA [51]. In patients with coexisting CA, AS severity should be assessed according to the current guidelines [52]. Approximately 50% of patients with confirmed CA have a severe low-flow low-gradient AS with preserved LV ejection fraction, (the so-called “paradoxical low-flow, low-gradient pattern”) [53], characterized by severe LV concentric remodeling, impairment of diastolic filling, left atrial remodeling and dysfunction, markedly reduced LV global longitudinal strain function, and right ventricular remodeling and dysfunction [4,25]. Additional imaging tests are required to differentiate a true-severe versus a pseudo-severe AS, such as dobutamine stress echocardiography noncontrast computed tomography in order to quantitate the aortic valve calcium burden. Until now, there is no randomized trial and no expert consensus that determines the best management of CA in patients with AS. There are very few data on the outcome and therapeutic management of patients with AS and concomitant CA. Most studies reported a high risk of mortality and nonimprovement in functional status following AVR in patients with severe AS and CA [53,54,55]. One study in a small number of patients (n < 30) suggests that the outcome of patients with AS and CA may be better with TAVR versus SAVR [53].

## 4. Conclusions

The cardio-oncology patient population has been increasing in recent years, requiring appropriate management strategies to improve quality of life and survival rate. The coexistence of significant aortic valve stenosis and cancer is relatively common and poses diagnostic and therapeutic dilemmas. Collaboration between cardiologists and oncologists is of primary importance to select the best treatment approach and optimize the timing of intervention. Further clinical trials and registry studies are needed to better appreciate outcomes in this complex setting. 

## Figures and Tables

**Figure 1 jcm-12-05804-f001:**
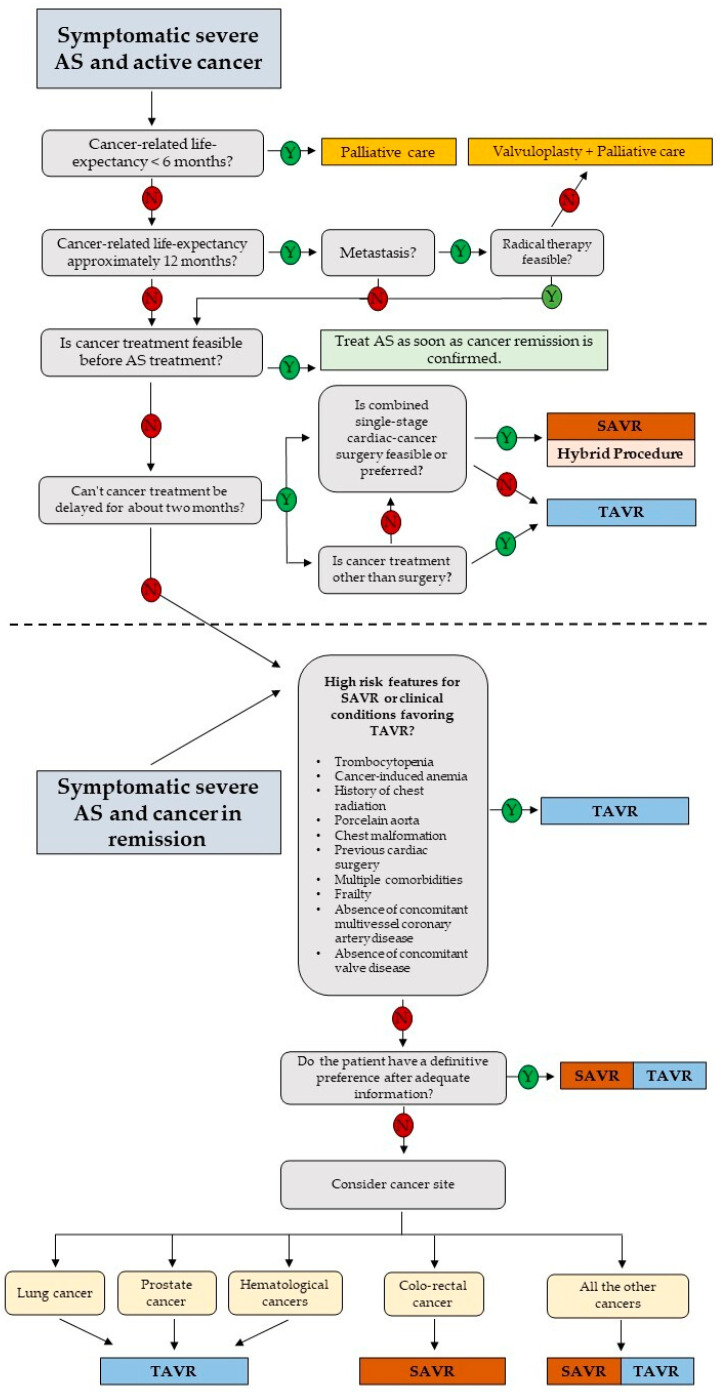
Proposed decision-making algorithm for the management of patients with severe aortic stenosis and cancer. Modified from ref. [34]. AS: aortic valve stenosis; SAVR: surgical aortic valve replacement; TAVR: transcatheter aortic valve replacement.

**Table 1 jcm-12-05804-t001:** Prevalence of cancer in patients with severe aortic stenosis according to available studies.

Author	Reference	All Population, n	Cancer, n (%)	Most Frequent Type of Tumor n, (%)
Faggiano et al.	[2]	240	64 (26.6%)	na
Mangner et al.	[3]	1821	99 (5.4%)	Prostate, 25 (25%)
Minamino-Muta et al.	[6]	3815	513 (13.4%)	na
Okura et al.	[5]	26,325	111 (0.42%)	Stomach, 13 (14.1%)
Guha et al.	[4]	47,295	27,960 (37.8%)	na

## Data Availability

Not applicable.

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
