# Peer review of "Aortic Valve Stenosis and Cancer: Problems of Management"

_jcm, 2023, doi:10.3390/jcm12185804_

Round 1

Reviewer 1 Report

Clear coverage of a very important problem: the co-occurrence of the most common acquired heart defect, which is aortic valve stenosis, and cancer-one of the most common diseases in the world. A growing population of people, requiring a look and separate randomized trials. 

Good points: comprehensive coverage of the problem

I suggest adding a few sentences about RT:

- the nature of valve calcification after radiotherapy (involvement of conjunctiva, involvement of subvalvular apparatus, etc.-anatomy inferior to TAVI, involvement of multiple valves simultaneously, 

- AS after RT, in case of mitral regurgitation after TAVI possibility of MitraClip

- After RT pulmonary fibrosis- worse ventilation with possible SAVR, worse healing of the sternum- skin fibrosis after RT

Please add the publication below about changes after RT:

Management of valvular heart disease in patients with cancer: Multidisciplinary team, cancer-therapy related cardiotoxicity, diagnosis, transcatheter intervention, and cardiac surgery. Expert opinion of the Association on Valvular Heart Disease, Association of Cardiovascular Interventions, and Working Group on Cardiac Surgery of the Polish Cardiac Society.

Płońska-Gościniak E, Piotrowski G, Wojakowski W, Gościniak P, Olszowska M, Lesiak M, Klotzka A, Grygier M, Deja M, Kasprzak JD, Kukulski T, Kosmala W, Suwalski P, Kolowca M, Widenka K, Hryniewiecki T.Kardiol Pol. 2023;81(1):82-101. doi: 10.33963/KP.a2023.0023. Epub 2023 Jan 15

Author Response

Dear Reviewers,

we would like to thank you for the helpful comments and suggestions. In the following, we outline the major changes that we have made to the manuscript to address the specific remarks. Your comments are in regular text and our responses follow in bold text

Reviewer#1

Clear coverage of a very important problem: the co-occurrence of the most common acquired heart defect, which is aortic valve stenosis, and cancer-one of the most common diseases in the world. A growing population of people, requiring a look and separate randomized trials. 

Good points: comprehensive coverage of the problem

Author: We would like to express our deepest thanks to the Reviewer’s valuable comments

Reviewer#1

I suggest adding a few sentences about RT:

- the nature of valve calcification after radiotherapy (involvement of conjunctiva, involvement of subvalvular apparatus, etc.-anatomy inferior to TAVI, involvement of multiple valves simultaneously, 

Author: Thanks to the Reviewer for the relevant observation. As suggested, we have added the following sentence:

Radiation-induced valvular disease involves a degenerative process  with early valve retraction resulting initially in regurgitation and after, thickening and calcification of valves leading to stenosis. Fibrosis and calcification of the mitral and aortic valves, especially surrounding structures including the annulus, subvalvular apparatus, and the aorto-mitral curtain (34), have been noted in patients who underwent C-XRT.  Left-sided valves most commonly affected probably due to the stimulus of higher pressure flows.

Reviewer#1

- AS after RT, in case of mitral regurgitation after TAVI possibility of MitraClip

Author

As we reported in the section "Aortic stenosis, coronary artery disease (CAD) and mitral regurgitation", at present in patients with severe MR, there is not enough experience to make recommendations about surgery versus combined or sequential TAVR and percutaneous mitral edge-to - edge repair. Currently no consistent data is available about the coexistence of AS and MR and the related treatment in the subgroup of cancer patients. In the general populations, patients with severe MR have been generally excluded from randomized clinical trials, making poor the impact that associated MR can have on clinical outcomes after TAVR. Incidence and prognostic impact of concomitant MR in patients undergoing TAVR requires specific attention as might trigger adjunctive strategy treatment which should be carefully evaluated in clinical trials.

Reviewer#1

- After RT pulmonary fibrosis- worse ventilation with possible SAVR, worse healing of the sternum- skin fibrosis after RT

Author:

Furthermore, C-XRT causes  mediastinal adhesions and fibrosis that need to be dissected. They increase the risk of bleeding and poor wound healing. The treatment with  debridement may increase the cardiopulmonary bypass time.

Author: Thanks to the Reviewer for the relevant observation. As suggested, we have added the following sentences:

Reviewer#1

Please add the publication below about changes after RT:

Management of valvular heart disease in patients with cancer: Multidisciplinary team, cancer-therapy related cardiotoxicity, diagnosis, transcatheter intervention, and cardiac surgery. Expert opinion of the Association on Valvular Heart Disease, Association of Cardiovascular Interventions, and Working Group on Cardiac Surgery of the Polish Cardiac Society.

Płońska-Gościniak E, Piotrowski G, Wojakowski W, Gościniak P, Olszowska M, Lesiak M, Klotzka A, Grygier M, Deja M, Kasprzak JD, Kukulski T, Kosmala W, Suwalski P, Kolowca M, Widenka K, Hryniewiecki T.Kardiol Pol. 2023;81(1):82-101. doi: 10.33963/KP.a2023.0023. Epub 2023 Jan 15

,

Author:

As requested, we added the following sentence and reference: As recently reported by Płońska-Gościniak, patients with severe, pre-existing cancer and heart valve disease should be managed according to the 2021 guidelines of the European Society of Cardiology and Cardiothoracic Surgery taking into consideration the cancer prognosis and patient preferences.

Reviewer 2 Report

Review of article: ‘Aortic valve stenosis and cancer: problems of management’ – J Clin Med

This review by Santangelo et al. analyzes the currently available Literature on the issue of aortic stenosis in patients with concomitant neoplastic diseases.

The coexistence of cancer and calcific aortic stenosis is quite frequent considering that patient age is increasing as the likelihood to present with valvular disease and various malignancies. Therefore, this review appears timely and well written since all various aspects of this issue are considered.

I would modify the table including for each article quoted the most frequent type of tumor observed in each series.

Although the authors do not provide substantial new information on this topic as they conclude that Further clinical trial and registry studies are needed to better appreciate outcomes in this complex setting’, nevertheless, this review should stimulate others to provide long-term data to improve current guidelines and verify the real indications in the management of patients with aortic stenosis and various types of malignancies.

Minor changes

Author Response

Dear Reviewers,

we would like to thank you for the helpful comments and suggestions. In the following, we outline the major changes that we have made to the manuscript to address the specific remarks. Your comments are in regular text and our responses follow in bold text

Reviewer#2

This review by Santangelo et al. analyzes the currently available Literature on the issue of aortic stenosis in patients with concomitant neoplastic diseases.

The coexistence of cancer and calcific aortic stenosis is quite frequent considering that patient age is increasing as the likelihood to present with valvular disease and various malignancies. Therefore, this review appears timely and well written since all various aspects of this issue are considered.

I would modify the table including for each article quoted the most frequent type of tumor observed in each series.

Although the authors do not provide substantial new information on this topic as they conclude that ‘Further clinical trial and registry studies are needed to better appreciate outcomes in this complex setting’, nevertheless, this review should stimulate others to provide long-term data to improve current guidelines and verify the real indications in the management of patients with aortic stenosis and various types of malignancies.

Author: We would like to express our deepest thanks to the Reviewer’s valuable comments

We added in table 1 for each article quoted the most frequent type of tumor observed in each series as follows:

Author

Reference

All population, n

Cancer, n (%)

Most frequent type of tumor n, (%)

Faggiano et al.

Int J Cardiol.2012;159(2):94-9

240

 64(26.6%)

na

Mangner et al.

J Interv Cardiol. 2018;31(2):188-96

1821

99(5.4%)

Prostate, 25 (25%)

Minamino-Muta et al.

Eur Heart J Qual Care Clin Outcomes. 2018;4(3):180-8.

3815

513 (13.4%)

na

Okura et al.

Int Heart J. 2018;59(4):750-8.

26325

111(0.42%)

Stomach, 13 (14.1%)

Guha et al.

J Am Heart Assoc. 2020;9(2):e014248

47295

27960 (37.8%)

na